Leadership and management influences the outcome of wildlife reintroduction programs: findings from the Sea Eagle Recovery Project

Sutton Alexandra E. Alexandra.Sutton@duke.edu
Nicholas School of the Environment, Duke University , Durham, NC , USA
Somers Michael
Electronic publication date: 2015 Jun 18
Publication date: 2015
Volume: 3
Electronic Location ID: e1012
Received 2014 Dec 15; Accepted 2015 May 19
Copyright: © 2015 Sutton
Copyright year: 2015
Copyright holder: Sutton
License: This is an open access article distributed under the terms of the Creative Commons Attribution License, which permits unrestricted use, distribution, reproduction and adaptation in any medium and for any purpose provided that it is properly attributed. For attribution, the original author(s), title, publication source (PeerJ) and either DOI or URL of the article must be cited.
License URL: https://creativecommons.org/licenses/by/4.0/

Keywords: Conservation leadership, Conservation champions, Transformational leadership, Wildlife reintroduction, White-tailed sea eagle, Haaliaeetus albicilla, Organizational culture

Funding: MSC L.T. Jordan Institute for International Awareness at Texas A&M University The Department of Wildlife & Fisheries at Texas A&M University Funding for this work was provided by the MSC L.T. Jordan Institute for International Awareness at Texas A&M University, and by the Department of Wildlife & Fisheries at Texas A&M University. The funders had no role in study design, data collection and analysis, decision to publish, or preparation of the manuscript.

==============================
Wildlife reintroductions and translocations are statistically unlikely to succeed. Nevertheless, they remain a critical part of conservation because they are the only way to actively restore a species into a habitat from which it has been extirpated. Past efforts to improve these practices have attributed the low success rate to failures in the biological knowledge (e.g., ignorance of social behavior, poor release site selection), or to the inherent challenges of reinstating a species into an area where threats have already driven it to local extinction. Such research presumes that the only way to improve reintroduction outcomes is through improved biological knowledge. This emphasis on biological solutions may have caused researchers to overlook the potential influence of other factors on reintroduction outcomes. I employed a grounded theory approach to study the leadership and management of a successful reintroduction program (the Sea Eagle Recovery Project in Scotland, UK) and identify four critical managerial elements that I theorize may have contributed to the successful outcome of this 50-year reintroduction. These elements are: 1. Leadership & Management: Small, dedicated team of accessible experts who provide strong political and scientific advocacy (“champions”) for the project. 2. Hierarchy & Autonomy: Hierarchical management structure that nevertheless permits high individual autonomy. 3. Goals & Evaluation: Formalized goal-setting and regular, critical evaluation of the project’s progress toward those goals. 4. Adaptive Public Relations: Adaptive outreach campaigns that are open, transparent, inclusive (esp. linguistically), and culturally relevant.

Introduction

Wildlife reintroductions are complex, expensive, and time-consuming. Worse, they are statistically unlikely to succeed, as repeated audits have shown (Clark & Westrum, 1989; Griffith et al., 1989; Kleiman, 1989; Fischer & Lindenmayer, 2000; Reading, Clark & Kellert, 2002; Lipsey & Child, 2007; Seddon, Armstrong & Maloney, 2007; Reading, Miller & Shepherdson, 2013). They are also the only way to restore an extirpated species to its prior home in cases where natural recolonization is impossible or unlikely, and for this reason, reintroductions remain an essential tool in conservation (Tear et al., 1993; Ostermann, Deforge & Edge, 2001).

Understanding success and failure in wildlife reintroductions

Much of the previous literature has attributed failures in reintroduction to deficiencies in the biological knowledge. Such theories presume that reintroduction outcomes are constrained only by the availability of biological data (e.g., Armstrong & Seddon, 2007; Cook, Morgan & Marshall, 2010). If this were the case, then reintroductions of data-rich species (e.g., wolves, lions) would be reliably more successful; they are not.

Some fault may lie in the inherent fragility of reintroduction procedures: the combined vulnerabilities of (i) small founding populations (Pimm, Jones & Diamond, 1988; Pimm, 1989); (ii) complex extinction causes (e.g., the passenger pigeon, which suffered simultaneously from overhunting, habitat loss, fragmentation of food landscapes, and lost cohesion of social groups (Bucher, 1992)); and (iii) potential loss of behavioral or genetic integrity due to captive breeding (Jule, Leaver & Lea, 2008) may prove insuperable in the re-establishment of an extirpated population.

Reintroduction is also relatively novel within the broader context of conservation—only within the past 40 years has it become a commonly-used scientific tool, and has had little time to form a body of knowledge about best practices to guide projects (Kleiman, Stanley Price & Beck, 1994; Sarrazin & Barbault, 1996; Stanley Price & Soorae, 2003; Seddon, Armstrong & Maloney, 2007; Robert et al., 2015).

It should come as no surprise, therefore, that most reintroductions fail. There has been some evidence that supplementary movements (such as the overwhelmingly successful (94%) grazing mammal translocations of South Africa, documented in Van Houtan et al., 2009) may flourish, but overall success rates remain low. Estimated rates of success vary between reviewers (46%—Griffith et al., 1989; 11%—Beck et al., 1994; 20% for restoration projects overall—Lockwood & Pimm , 1999; 26%—Fischer & Lindenmayer, 2000; 53% for wild-born carnivores, 32% for captive-born—Jule, Leaver & Lea, 2008), but the pattern remains clear: in recreating an absent population, some efforts succeed; most do not.

Understanding success and failure in organizations

Organizations, likewise, may succeed but often fail. This failure can be linked strongly to the organization’s internal activity: the set of behaviors and values that establish professional norms and direct operations within an institution. This set of behaviors and values has been termed organizational culture, and has been under study since the early 1980s in the business and management research fields (see: Schein, 1984).

An organization’s culture manifests in every aspect of the institution, including such structures as administrative hierarchies, staff competencies and experience, financial resources, and management practices (Schein, 1990; Schein, 2010; Lunenburg, 2011). Expectations about each of these inform and restrict decision-making within an organization, and in doing so, culture becomes directly influential on outcomes (Barney, 1986; Schein, 1990; Schein, 2010). This is a complex explanation for an intuitive phenomenon: that a well-run organization will perform better than a poorly-run one.

Despite conservation’s origins in scientific practice, it is fundamentally an applied field, and as such, relies on practice and operation to achieve desired outcomes. In this sense, a conservation initiative, entity, or project does not differ from other organizations, and is just as subject to the influence and impact of human and organizational factors. In fact, organizational experience, preference, and priorities direct every decision about reintroduction from the first recognition of the loss of a species. Biases towards charismatic species, cultural preferences, the geopolitical context of reintroduction, the depth of existing scientific knowledge, and questions of physical accessibility all shape projects in their planning phases. Organizational structures, staff selection and experience, leadership and management styles, funding availability, and cultural identity all shape projects throughout their working phases. Professional status, disciplinary culture, publication bias, and funding availability or obligations all influence projects in their monitoring phrases. So why have these areas been understudied?

Understanding wildlife reintroduction outcome as organizational performance

Past reviews of reintroduction outcomes have focused almost exclusively on identifying broad, biological prerequisites for success (Morris, 1986; Kleiman, 1989; Wolf et al., 1996; Sarrazin & Barbault, 1996; Wolf, Garland & Griffith, 1998; Fischer & Lindenmayer, 2000; Stanley Price & Soorae, 2003), limiting focus on the potential influence of human and organizational (i.e., human dimensions) factors (O’Rourke, 2014). Some attention has been paid to the issue of bias in species selection for reintroduction (Seddon & Van Heezik, 2013; Bajomi et al., 2010; Seddon, Soorae & Launay, 2005), but these studies are few and recent, and comprise only a small portion of the overall literature.

Leadership and day-to-day management, for example, form the foundation of any reintroduction program. Yet they are discussed sparingly in the general discourse, and very few places discuss them in the early literature: only Morris (1986) and Kleiman (1989) acknowledge the necessity of engaging with the public and obtaining the governmental support. Reading & Miller’s (1994) chapter expressed some interest in organization and management: “Endangered species recovery programs could be greatly improved by addressing their professional and organizational weakness.” (p. 73), and a brief (but skeptical) acknowledgment exists in Wolf et al.’s (1996) paper: “Although management techniques are not applied uniformly among translocation programs …little relevant data exist to indicate whether this was an important issue.” (p. 1150). Other contemporary researchers continued to downplay the potential impacts of these non-biological factors, arguing instead that demography, genetics, and ecology were the truly decisive influences on success (Sarrazin & Barbault, 1996).

Reading, Clark & Griffith returned to the topic in 1997, but the researchers used a mailed questionnaire approach that provided data too coarse to link specific aspects of leadership and management (in their terms: ‘valuational and organizational considerations’) to program outcomes. Miller et al. touched momentarily on the issue again in 1999: “A well-trained and dedicated staff with the appropriate expertise is crucial to program success …. For that reason, careful attention to the organizational structure of the decision-making body is crucial to maintaining an efficient and effective program”, (p.65) but subsequent studies did not further pursue this suggestion. And although Beck made overtures toward this in his introduction to a special issue of the Association of Zoo & Aquarium’s Communiqué in 2001, saying “…reintroduction is as much a sociological, political and economic undertaking as it is biological,” attention to the topic remained limited thereafter.

In the last year, three publications have significantly advanced the dialogue on leadership and management as pertains to reintroductions:

Post & Pandav’s (2013) review of tiger reserves (where several reintroductions have taken place) in India highlighted the criticality of leadership, finding that “the presence of ‘conservation champions can dramatically affect the performance of individual reserves.” (‘Champions’ were first defined by Andersson & Bateman in 2000 as ‘Individuals who …possess environmental knowledge and skills (that) are key factors in the mobilization of support.’)

O’Rourke’s (2014) case study of the reintroduction of the white-tailed sea eagle to Ireland encouraged several management shifts for future projects (greater engagement in stakeholder dialogue, increased emphasis on the human dimensions of reintroductions, and adoption of a holistic, interdisciplinary approach to future projects) and concludes, “The reintroduction of a species into its former range is only partly about biology—socio-economics, politics and social acceptability (are) equally important” (p. 135).

And last, but hardly least: the International Union for Conservation of Nature (IUCN) has released an updated (2013) version of its Reintroduction Guidelines. The guidelines revisit many of the general recommendations from the original document, but expound further on some related to our topic, most particularly in section 4.1 (“Goals, objectives, and actions”); 5.2 (“Social feasibility”); 8.1 (“Social, cultural and economic monitoring”); and in Annexes 2.5, 3.1.14, and 6.3.5 (Definitions, Deciding When, and Risk Analysis).

Each of these provides valuable support for increased emphasis on understanding the impact of human dimensions on reintroduction outcomes, but none delve deeply into the internal organizational factors that might support or detract from potential success.

My study augments the findings of previous researchers with an in-depth exploration of the impact of both human dimensions and organizational factors on the success of a high-risk reintroduction program: the Sea Eagle Recovery Project, which took place from 1975–2012 in Scotland.

A brief history of sea eagles

The white-tailed sea eagle (Haaliaeetus albicilla), in the family Accipitridae, is the largest bird of prey in the United Kingdom (Fig. 1). It possesses a wingspan over 2 m, and an average male/female weight of 4.5/6 kg, with females significantly larger than males (Love, 1983; Royal Society for the Protection of Birds, 2006). Adults of the species are brown with pale heads and white, wedge-shaped tails, yellow beaks, yellow un-feathered legs, and golden eyes (Love, 1983; Royal Society for the Protection of Birds, 2006). The white-tailed sea eagle’s (hereafter, “sea eagle”) range extends over most of northern Europe and Asia, with roaming birds observed as far south as the Mediterranean (Royal Society for the Protection of Birds, 2006). The eagles further have a long history in Scotland, with referent placenames dated as early as 500 CE (Evans, O’Toole & Whitfield, 2012) and representations appearing in Pictish carvings predating the Stone Age (Love, 1983). The diet of the eagle consists primarily of fish and small mammals, with occasional predation of small birds and scavenging of carrion.

Figure 1 Sea eagle, pre-release, on its nest in captivity in Scotland, 2009.

Extinction

White-tailed sea eagles (Haaliaeetus albicilla) were large, bold birds that quickly habituated to humans, dined on managed grouse, and predated lambs; they were therefore intolerable pests to British gamekeepers and crofters of the 19th century (Love, 1983; Lister-Kaye, 1994; Royal Society for the Protection of Birds, 2005). Further, sea eagle specimens became a favorite of Victorian egg collectors, and traders regularly raided the birds’ nests (Love, 1983). The sea eagle thereby began to decline in the 19th century, and was extinct in Britain by the early 20th. The last wild pair were on the Isle of Skye in 1916, and the last wild individual was shot in Shetland in 1918 (Baxter & Rintoul, 1953; Love, 1983; Mudge et al., 1996; Bainbridge et al., 2002).

When the sea eagle reintroduction began in 1975, the project faced major challenges that put it at high risk for a lack of success:

Ongoing land use conflict

Significant changes had taken place in the British economy, wildlife laws, and gamekeeping practices since sea eagles were extirpated in 1918, suggesting that the original threats to the birds had likely diminished so far as to be negligible by the mid-1970s. However, contemporaneous studies of the golden eagle (Aquila chrysaetos) revealed ongoing challenges with persecution, habitat loss, and disturbance (e.g., Newton, 1972).

Experimental Failure. Two pilot reintroduction attempts were made in 1959 and 1968 (Sandeman, 1965; Dennis, 1969; Green, Pienkowski & Love, 1996), but by 1975, when the official reintroduction began, not a single bird had reestablished in Scotland.

Limited biological knowledge

In 1975, no body of knowledge about the process of reintroduction existed upon which project members might have based their work. Although the eagle was plentiful in Norway, scientists knew little about its ecology in Scotland (Love, 1979). Bird reintroductions are, as a whole, less successful than mammalian projects (Wolf et al., 1996), and carnivores less than omnivores (Wolf, Garland & Griffith, 1998). Raptor reintroductions are thus doubly cursed, and although overrepresented as a percentage of bird reintroductions (Seddon, Soorae & Launay, 2005), are more likely to fail.

Lack of government support

The Wildlife & Countryside Act of 1981 established clear guidelines for the importation and release of native species into the United Kingdom, but prior limitations set by the Animals (Restriction of Importation) Act of 1964 had already established a precedent of strictly avoiding the importation of any animal to the country. Morris (1986) notes that even after the 1981 Act granted greater license, a strong fear of unintentionally harmful introductions persisted. And since such a large-scale bird project had no precedent at that time in Britain, support for such a risky—if pioneering—project was limited, hard-won and tentative (Tingay & Katzner, 2012).

Conclusion & success

From 1975–2012, the Sea Eagle Recovery Project released 167 juvenile birds, resulting in 350+ adult animals and 65+ breeding pairs across Scotland (Smith, 2007; Patterson, 2010; Scottish Natural Heritage, 2014). Releases between 1975 and 1998 resulted in 42 territorial pairs (Evans et al., 2009; Hipfner et al., 2012), rising to 44 territorial pairs by 2008/9 (Sea Eagle Project Team, 2008; Grant, Reid & Whitfield, 2011) and 79+ territorial pairs by 2013 (Scottish Natural Heritage, 2014). By the Project’s conclusion, the popular media (Public Broadcasting Service, 2010; British Broadcasting Company, 2013), conservation literature (Whitfield et al., 2009; van Wieren, 2012), and government leaders (Scottish Natural Heritage, 2014; National Farmers Union of Scotland, 2014) all agreed that the project had been a success.

In the study presented here, I explore some of the ways in which human and organizational factors (specifically: leadership and management) of the recovery project may have contributed to this successful outcome.

Methods

I drew on data from multiple sources—interviews, observations, archival records, publicity documents, scientific publications, internal reports, and multimedia materials—as well as two traditions of inquiry: the case study and grounded theory methods. This approach relied on interviews with human subjects, and was approved by the Texas A&M University Institutional Review Board under IRB Protocol #20080131.

Selection of focal project

I chose the Sea Eagle Recovery Project because of its length (>40 years), status at the time of research (ongoing), success, and relative celebrity within the country (Scottish Natural Heritage, 1995; Royal Society for the Protection of Birds, 2006; Scottish Natural Heritage, 2007; Scottish Natural Heritage, 2008; British Broadcasting Company, 2008; Evans et al., 2009). Of further benefit was the fact that the reintroduction took place in four discrete phases: a pilot study in Fair Isle, the first phase in the Inner Hebrides, the second in Western Scotland, and the third in Eastern Scotland. These discrete phases allowed me to compare shifts in leadership and management across the length of the project, providing a natural experiment that gave insight into how different approaches might have influenced outcomes.

Data collection

I conducted face-to-face, in-depth, semi-structured confidential interviews with verbally consenting, voluntary participants who had been full-time project employees for at least three months during any phase of the reintroduction program. I asked about individual interviewee’s experience with sea eagles during, before, and after the reintroduction, as well as the organizational structure of the project during the individual’s time of employment, and the overall experience of working with the project (for a full list of guiding questions, see Appendix S1). I also asked interviewees to recommend other potential interviewees (the “snowball method”; Goodman, 1961).

In interviews, I made use of a modified logic model framework, based in the Gugiu & Rodriguez-Campos semi-structured interview protocol (2007), to guide the interview process. This method consisted of a series of introductory questions which ask basic information about the interviewee, followed by a series of open-ended questions intended to encourage the speaker to speak freely about their experiences. I set no time limit for the interviews. This approach allowed me to collect detailed accounts of the program and work in-depth with my interviewees to gain an understanding of organizational culture (Lincoln & Guba, 1985; Erlandson, 1993).

I conducted interviews with 13 interviewees in various locations (convenient to the interviewee) across Scotland, but eliminated two candidates post hoc. This is because one interviewee turned out to have worked for less than three months on the reintroduction (and therefore did not meet the criteria for inclusion), and because one interviewee’s recordings were entirely lost due to technical failure.

I therefore conducted 17 total interviews, but after two eliminations, only 15 of these were ultimately used. I also conducted follow-up interviews via Skype with four of the six most experienced interviewees (those who had worked through at least two phases of the reintroduction); two were excluded because of schedule unavailability.

In addition to interviews, I gathered documents including but not limited to public outreach papers and pamphlets, children’s education books, curricular materials, internal and external newsletters, newspaper and internet articles, blog posts, books, informational and recruitment brochures, DVDs, recorded TV programs, community flyers, and other informational packets either presented by or related to the project. I collected these items from archival collections at the Royal Society for the Protection of Birds (RSPB) Scotland headquarters, the Scottish Natural Heritage (SNH) offices, a variety of wildlife centers located around the country, and from private collections.

Data analysis

Manual typology

Extracting useful information from qualitative data first necessitates organizing the collected data into discrete groups or categories (Caracelli & Greene, 1993; Stake, 1995; Creswell, 2007). I began by grouping my interviews, documents, and notes into broad, meaningful types (e.g., children’s books; brochures; journal articles; scientist interviews; non-academic texts). I then read and analyzed each document, identifying and highlighting (“tagging”) recurrent concepts to create a preliminary data chart (“typology”) (Caracelli & Greene, 1993; Creswell, 2007). As I read, I tagged discrete and overlapping passages, words, or phrases that described a particular thought, idea, or concept. This process matches the overall approach that both Stake (1995) and Creswell (2007) suggest for conducting either grounded theory or traditional case study research.

My tagged and highlighted passages resulted in an initial list of over 57 discrete ideas, concepts, and experiences; I then grouped these discrete experiences into a shorter list of eight categories (see: Experience Type Codes, Table 1). I then tagged discrete, descriptive characteristics within each Type (e.g. ‘It was really quite helpful having our supervisor around a lot.’ would have been categorized as Contact with Supervisor/Frequent/Positive; see Experience Characteristic Codes, Table 1).

Table 1 Management themes and characteristics of the Sea Eagle Recovery Project. Definitions of Selected Terms.

Autonomy refers to the ability of team members to complete their work independently, while either in the office or in the field. Hierarchy refers to the assignation of responsibilities and privileges to team members according to a graded or ranked system. Accountability refers to the ability or expectation of practitioners to explain or justify their actions through formal or informal evaluation or review. Evaluation refers to the complete process of professional assessment, which may take place under the authority of either internal or external entities. Public Relations/Outreach refers to the effort made by the project to interact with, access, educate, or include members of the public during the reintroduction process.

Experience Type (ET) codes	Descriptive Experience Characteristic (EC) codes	
Contact with Supervisor (CS-)	Frequent (F) ‖ Infrequent (I)	
	Positive (+) ‖ Negative ($) ‖ Neutral (N)	
Position/Job Duties (JD-)	Autonomous (A) ‖ Non-autonomous (Na)	
	Primary (P) ‖ Secondary (S)	
	-Fieldwork (Fw)	
	-Administrative work (Aw)	
	-Public Relations work (PRw)	
	-Supervision of Others (So)	
Relationship with Coworkers (RC-)	Shared Responsibilities (SR) ‖ Divided Responsibilities (DR)	
	Egalitarian (E) ‖ Hierarchical (H)	
Goal-Setting and Evaluation Process (GSE-)	Proximate (P) ‖ Ultimate (U)	
	- Formal (L) ‖ Informal/Casual(C)	
	- Beneficial (+) ‖ Unhelpful/Costly ($)‖ Neutral (N)	
	- Frequent (F) ‖ Infrequent (I)	
Contact with Public (CP-)	Positive (+) ‖ Negative ($) ‖ Neutral (N)	
	Frequent (F) ‖ Infrequent (I)	
Public/Media Relations (PR-)	Internally Generated (Y)‖ Externally Generated (X)	
	-Positive (+) ‖ Negative ($) ‖ Neutral (N)	
	-Frequent (F) ‖ Infrequent (I)	
Program Progress (PP-)	Good (G) ‖ Poor/Bad (B) ‖ Neutral (N)	
Program Performance (PO-)	Good (G) ‖ Poor/Bad (B) ‖ Neutral (N)	

Once I completed this process for all of my collected documents, interviews, multimedia, and texts, I created a final data chart encompassing all the concepts, their characteristics, and the strength of their recurrence across multiple data sources. The typology I extracted from that final data chart is presented in Table 1.

Digital typology

After the construction of a manual typology, I imported all interviews and digital documents into NVivo 10, a qualitative analysis software program, and then used the manual typology as a guideline for inductive digital analysis. This approach afforded me the opportunity to code more precisely and to explore the data with greater nuance, including queries and cross-tabulations of thematic overlap (Auld et al., 2007; NVivo, 2013).

Results

Interviews averaged 45 min, and all took place at times and locations of the interviewee’s choice.

Interviewee demographics

Interviewees had worked an average of 18.3 years on the Sea Eagle Recovery Project, and had lived in Scotland an average of 30.8 years (more than half of interviewees were lifelong residents of Scotland). Six interviewees had worked through more than one phase of the reintroduction; four had served during the earliest phases of the project (1968–1990) and ten had served during the latter phases of the project (1990 onward). Nine of eleven interviewees were men (Table 2).

Table 2 Demographics of Interviewees within the Sea Eagle Recovery Project.

Gender	Employer during Sea Eagle Recovery Project	Length of time living in Scotland	Years working with Sea Eagle Recovery Project	Phases* involved	
M	RSPB	40 years	41	All	
M	SNH	20 years	19	2 + 3	
M	SNH	Whole life	19	2 + 3	
M	RSPB	20 years	8	1 + 2	
M	RSPB	Whole life	1	2 + 3	
M	SNH	Whole life	10	2 + 3	
M	Several	Whole life	41	All	
F	RSPB	Whole life	15	2 + 3	
M	SNH	5 years	25	2 + 3	
M	RSPB	20 years	25	1, 2, 3	
F	RSPB	4 years	2	3	
Notes.

* Phases refer to the following: 1959—Pilot Phase (Fair Isle) 1975–1985—Phase 1: the Hebrides (Isle of Rum) 1993–1998—Phase 2: Western Scotland (Wester Ross) 2007–2012—Phase 3: Eastern Scotland (Fife).

Most were currently employed by the Royal Society for the Protection of Birds (n = 4) or Scottish Natural Heritage (n = 3); one interviewee was employed by Forestry Commission Scotland; and the remainder (n = 3) were self-employed. During their work on the reintroduction, six of the 11 interviewees had been employed by the Royal Society for the Protection of Birds, the majority remainder (n = 4) had been employed by Scottish Natural Heritage. One interviewee had been employed by multiple organizations, beginning with the Nature Conservancy Council.

Interview summary

Interviewees referenced a number of recurrent human and organizational issues that may have been influential to project outcomes, comprising four overall experience themes, which are highlighted below:

Theme 1: Leadership/Management, Hierarchy & Autonomy

Theme 2: Goals, Targets & Evaluation

Theme 3: Public Relations/Community Outreach

Theme 1: leadership & management, hierarchy & autonomy

More than half of interviewees’ total reports on the nature of their experience described contact with supervisors as infrequent (n = 4, 57%) but positive (n = 4, 57%). These reports were made concurrent with verbal and nonverbal expressions of neutrality. More than half of interviewees described their work as autonomous (n = 6; 54.5% of respondents) and all interviewees could clearly identify their own supervisors and key project advisors, as well as accurately detail the chain of command above and below them (n = 11; 100% of respondents). Most interviewees’ reports described the structure of their program as hierarchical (n = 45, 51.72%). Most reports on the nature of work within the reintroduction also described specialized assignments and clear task division between employees (n = 43, 65%). Early phase participants reported slightly less hierarchy and greater autonomy than later-phase participants, but the difference was marginal, and overall descriptions were consistent throughout reintroduction phases (Fig. 2).

Figure 2 Consistency in describing the nature of work in the Sea Eagle Recovery Project across phases, as determined by frequency-of-mention in a digitized typological analysis using NVivo software.

Phases refer to the following: 1959—Pilot Phase (Fair Isle) 1975–1985 —Phase 1: the Hebrides (Isle of Rum) 1993–1998—Phase 2: Western Scotland (Wester Ross) 2007–2012—Phase 3: Eastern Scotland (Fife)

Theme 2: goals, targets & evaluation

Interviewee reports on the nature of goal-setting differed by phase, with Pilot Phase (1968) reports tending to describe the goal-setting process as infrequent (n = 3, 100% of reports) and ad hoc (n = 4, 100% of reports) while Official Phases (1975–2012) reports tended to describe the process consistently as infrequent (n = 6, 100% of reports) but formal and bureaucratic (n = 30, 94% of reports).

The frequency with which interviewees discussed the impact of long-term goal setting increased with the project’s progression, with the organizational influence of goal-setting arising four times more frequently with reference to the last phase of the project than the first (Pilot Phase frequency—1; Phase 1 frequency—1.75; Phase 2 frequency—3.28; Phase 3 frequency—4).

Evaluation likewise was discussed more frequently as influential to success in the latter phases of the project (Pilot Phase—1.75; Phase 1—2.75; Phase 2—3.29; Phase 3—3.71). Descriptive reports of the nature of evaluation were consistent across phases: evaluation within the project was generally formal (n = 27, 77% of reports), took place on an ongoing or ad hoc basis (n = 20, 67% of reports), and was handled internally (i.e., did not involve an external agency or auditor; n = 10, 100% of reports) (Fig. 3).

Figure 3 Demonstrating consistency in the nature of evaluation throughout the Sea Eagle Recovery Program, as determined by frequency-of-mention in a digitized typological analysis using NViVo software.

Phases refer to the following: 1959—Pilot Phase (Fair Isle) 1975–1985—Phase 1: the Hebrides (Isle of Rum) 1993–1998—Phase 2: Western Scotland (Wester Ross) 2007–2012— Phase 3: Eastern Scotland (Fife).

Theme 3: public relations & community outreach

Conflict and Persecution was by far the most frequently reported Public Relations issue (n = 102 reports), nearly doubling in frequency-of-mention between the first and last phases of the project (Phase 1 frequency: 3.25; Phase 4 frequency—5.28) across all four phases of the project. Tourism was a distant second in frequency of discussion (n = 12 reports). Concurrent with interviewees’ reports of conflict and persecution were verbal and nonverbal expressions of feelings of frustration, sadness, anger, and/or resignation/fatigue (Fig. 4).

Figure 4 A word tree demonstrating the contextual mentions of ‘persecution’ by interviewees of the Sea Eagle Recovery Project, as determined from a query made in NVivo software as part of a digital typographical analysis.

This word tree provides some examples of the contextual language surrounding discussions of wildlife persecution in the Sea Eagle Recovery Project.

Discussion

Four critical factors in the human and organizational foundation of the Sea Eagle Recovery Project contributed to its success, helping it to overcome the challenges of limited biological knowledge, poor early support, and failures in its experimental pilot. These four critical success factors are common to all reintroduction projects, and the manner in which the Sea Eagle Recovery Project executed them could serve as an example for wildlife reintroductions worldwide.

Leadership & management

A small, dedicated team of experts who served as strong scientific leaders in addition to political advocates provided a huge boon to the project (as first suggested in Clark & Westrum, 1989). Roy Dennis and John Love invested huge amounts of time and personal capital in the first two decades of the Sea Eagle Recovery Project; their activities included everything from personally releasing the birds to giving testimony to local and national governance in support of more supportive wildlife laws.

Roy Dennis had already been working in the highlands of Scotland for nearly a decade and was the director of the Fair Isle Bird Observatory when he began work on this project. By chance, his 1968 trial release of four birds coincided with a visit to the bird observatory by John Love, a zoology undergraduate from the University of Aberdeen (Love, 1983; Love, 2006; Tingay & Katzner, 2012). By the time the project officially began in 1975, Dennis and Love had been working on re-establishing the bird for more than sixteen years. Love & Dennis became the senior leaders of the program, and while they recruited other scientists and experts to work with them, they maintained executive control over the project. This lent the project a sense of continuity and set a structure that (in combination with ongoing evaluation) buttressed the reintroduction against internal negligence. Without long-term, consistent leadership of this nature, it is unlikely that the reintroduction would have overcome its initial challenges.

This ‘champion’-style leadership (Andersson & Bateman, 2000; Post & Pandav, 2013) is the most consistent and perhaps most important advantage that the Project enjoyed, and was evident through all four phases of the reintroduction. This style of leadership fits into a larger categorization of ethical and transformational leadership—a style known to support positive organizational outcomes and guide employee attitudes with minimal interference in day-to-day employee operations (Toor & Ofori, 2009). This minimal interference is reflected in the infrequency/positivity of interviewees’ reports.

Hierarchy & autonomy

Positive contact with leadership and operation within a hierarchical framework (i.e., clear chains of command; assigned roles differentially by rank, etc.) improved employee morale and productivity by raising individual accountability and allowing a high degree of autonomy in completing those tasks. This management approach was well suited to both the specific needs of reintroduction projects (i.e., quick, decisive, responsive action in the field) and the desires of its participants (i.e., freedom to self-direct throughout the day), leading to marked efficiency.

The business literature suggests that autonomy confers significant benefits to performance in the presence of high-variety tasks, or when task interdependence within a group is high (Dodd & Ganster, 1996; Langfred, 2000). This has direct relevance for conservation programs, in which employees work as part of a team, must perform varied tasks competently, and must respond quickly and independently to changing conditions (Soulé, 1985; Clark & Westrum, 1989). Retaining high autonomy—even within a strict hierarchical structure—thus likely confers useful benefit to conservation practitioners.

Sea Eagle Recovery Project employees had a unique flexibility to take independent action when necessary, but also to ‘fall in’ to a known and clearly-defined hierarchy when expert assistance (provided by strong, dedicated leader-experts) was needed; this was yet another benefit conferred on the Project by its organizational culture which may have contributed to its success.

Goal-setting & evaluation

Scrutiny surrounding the advent of the Sea Eagle Recovery Project meant that Dennis, Love, and other project managers were under pressure to demonstrate clear, measurable success. This came initially in the form of annual reports on bird release numbers, rate of establishment, cost per bird, etc. These early reports were the precursors to the more formalized reporting system established by the Joint Nature Conservancy Council in the later Western phase.

Ongoing, critical internal evaluation (for an early advocacy of this method, see: Kleiman et al., 1999) strengthened the validity of the project’s practices and improved support among supporting entities (e.g. the Joint Nature Conservancy Council, Scottish Natural Heritage). The amount of accountability in an organization may reflect in its performance rating and evaluation process. Theoretically, the implementation of performance ratings increases accountability by holding participants responsible for actions taken and results produced. In reality, this may not always be the case, as performance ratings and evaluations may be inefficient, inappropriate, or counterproductive to improving performance (Halachmi, 2002; de Lancer Julnes, 2006; Tilbury , 2006).

Indeed, certain interviewees reported increasing concerns about the potentially negative impact of goal-setting and evaluation (“But I worry nowadays that they’re becoming too structured; that there’s just too many goals, that …some of it has become unnecessarily bureaucratic.”—Interviewee #13, 2009); this warranted further inquiry. An analysis of coding similarity using Jaccard’s coefficient confirmed that these interviewees were outliers; they had participated in the Pilot Phase of the project, a time during which formal evaluation of any kind was close to none, perhaps making them more aware of later changes in guidelines and evaluation of the project.

Overall, the clear goalposts and regular (if infrequent) evaluation of progress conferred yet another benefit on the Sea Eagle Recovery Project. This is in part because the establishment and evaluation of goals requires good organizational governance (e.g., clear structure and diligent leadership) as a pre-existing condition for efficacy; in this way, these three elements are woven into a framework to build success, and the sea eagle reintroduction was fortunate to possess them.

Public relations & conflict

It can be difficult to parse the contribution of public relations to the ultimate performance of an organization or project. This is because the intangible benefits of improved relationships, improved legitimacy, or improved public opinion can be difficult or cumbersome to measure (Bennett & Gabriel, 2001; Likely, Rockland & Weiner, 2006; Phillips, 2006). Wildlife reintroduction programs are uniquely interrelated with issues of public sentiment (Clark & Westrum, 1989; Kleiman, 1989; Seddon, Armstrong & Maloney, 2007). Thus, the likely relationship between public relations and program performance has definite salience to this field.

Indeed, incidents of persecution and conflict, particularly with local crofters and fishermen marred the earliest phases of the sea eagle reintroduction. Unexpectedly, the project had to contend with this onslaught of human-wildlife conflict. By the end of 2004, 25% of eagle mortality was attributable to persecution (Joint Nature Conservation Committee, 1988; Love, 2006). The trauma of these events weighed heavily on the project and its participants, making it the most-often cited public relations issue across all interviews, with 85 references made by 10 of the 11 interviewees (“Persecution is a major problem that some hard-line people will never give up—poisoning, especially—and that’s when sea eagles become vulnerable. But hopefully …the new generation will be better educated.”—Interviewee #7, 2009).

This early experience laid the painful paving stones for later shifts in the public relations strategy, however, and these shifts may have benefited the reintroduction—and the eagles—overall. The adaptive public approach that Project leaders eventually adopted reflected a growing understanding of the value of cultural sensitivity, inclusivity, transparency, and local “ownership” of conservation initiatives (for an example of unsuccessful implementation of this strategy in Ireland, see: O’Rourke, 2014). Shifting the discourse with the public toward scientific openness, direct address of complications and problems, improved linguistic parity, and linking the reintroduction to the public’s regional identity were likely key to engendering better support and eventually allowing the Project to succeed:

“We had two clutches of eggs stolen in one year and some local residents said, ‘Why didn’t you ask us to help watch the nest?’ So, we did. And it worked quite well. People have to, you know, get really involved and to feel that they are making a contribution. And it gave a sense of some importance in the community. Had we not done that, and sort of persisted in doing things the way we were, we’d be running the risk of saying, ‘Well, actually, these aren’t your birds at all. They are our birds. ‘Keep away from them.’ And that’s really the wrong attitude to take.”—Interviewee #11, 2009

This adaptive public relations strategy, begun as a reaction to conflict, became a meaningful and significant element of the Project’s organizational culture, and yet another contributing factor in the reintroduction’s success (for further discourse analysis, see: Arts, Fischer & Van der Wal, 2012).

Management Recommendations

Although these findings are limited by their exploratory (and therefore preliminary) nature, I draw on them to suggest four recommendations about best practices for organizational management in wildlife reintroduction projects:

1. Leadership & management: Reintroductions benefit from dedicated, consistent, long-term ‘champion-style’ leadership.

2. Autonomy & hierarchy: Reintroductions benefit from a clear hierarchical framework that serves as support for high employee autonomy in the field.

3. Goal-setting & evaluation: Reintroductions benefit from consistent, regular evaluation of progress toward formally established goals.

4. Public relations & outreach: Reintroductions benefit from adaptive public relations strategies that are open, transparent, inclusive (esp. linguistically), and culturally relevant.

Conclusion

The potential value of examining the conservation initiative (in this case, the reintroduction program) as an organization has been deeply neglected in the conservation literature. Despite its exploratory nature, the findings of this study suggest a specific and potentially fruitful direction which future research could take. Following studies could examine, broadly and comparatively, the differential outcomes of conservation initiatives with differing leadership and management styles. Such a comparative study would be a useful contribution to the growing wealth of literature related to conservation leadership and management.

Supplemental Information

Appendix S1 Semi-Structured Interview Survey Protocol (with Informed Consent Statement)

Click here for additional data file.

Supplemental Information 2 Raw data

Click here for additional data file.

Thanks to Drs R Lopez and SL Pimm, for guidance and support. Thanks to the staff of the Royal Society for the Protection of Birds, Scottish Natural Heritage, the Forestry Commission, and the Sea Eagle Recovery Project for their participation in interviews and contributions to my analysis. Thanks especially to K Duffy, for his help with facilitation and support. Thanks also to Phillip Seddon and Eileen O’Rourke, whose reviews greatly improved this manuscript.

Additional Information and Declarations

Competing Interests

Author Contributions

Human Ethics

The author declares there are no competing interests.

Alexandra E. Sutton conceived and designed the experiments, performed the experiments, analyzed the data, contributed reagents/materials/analysis tools, wrote the paper, prepared figures and/or tables, reviewed drafts of the paper.

The following information was supplied relating to ethical approvals (i.e., approving body and any reference numbers):

Texas A&M University

Institutional Review Board

Protocol for Human Subjects in Research

IRB Protocol #20080131.

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
