# Peer review of "Leadership and management influences the outcome of wildlife reintroduction programs: findings from the Sea Eagle Recovery Project"

_PeerJ, doi:10.7717/peerj.1012_

## Round 0.1 · original submission · Major Revisions

Like referee #2 I feel there is some important previous work ignored in this MS. If you sort out this issue by bringing in the latest IUCN guidelines and the other work suggested, and consider all the other comments by both referees I will consider a revision. I will send the revision back to both referees.

·

Basic reporting

No comments

Experimental design

This ms sits more in the Social Sciences than in the Biological, using semi-structured interviews to draw conclusions about the importance of management structures and processes. In addition, the limited sample size and the acknowledged preliminary nature of the work places the ms more as an opinion piece than as primary quantitative research.

Validity of the findings

This opinion article takes as its base the documented low success rate of wildlife reintroductions to make a case that to date the emphasis has been on the biological drivers of population establishment success, and as a consequence the importance of project leadership and management has been overlooked. Sutton uses the case study of sea eagles in Scotland to propose four managerial elements that he considers to be critical to project success.

A total of only 15 (L204) (or 11?, L247) people were interviewed using a semi-structured approach. Interviewees were asked to consider issues that might have been influential to success, thus a subjective assessment is being made by people, in isolation from considering the relative role that biological factors played in outcomes. As written in the Methods L259+ it seems interviews were structured around four themes (confusingly labelled as themes 1-3). Unsurprisingly therefore, the Discussion starts with the conclusion that (L297) these “four critical factors…contributed to success”. This seems like a tautology, whereby interviews have been structured around an a priori thesis of what themes are important, leaving no room to refute the hypothesis that these four (3?) themes are critical, no way to discover new themes, and no way to eliminate any of the 3-4 themes discussed.

One of weaknesses of this ms is a failure to consult the latest IUCN guidelines on reintroductions (2012), where considerable emphasis is indeed placed on at least 3 of the four the social aspects Sutton considers to have been overlooked:

1. Leadership and Management (ref IUCN 2012, Section 5.2)
3. Goals and Evaluation (ref IUCN 2012, Section 4.1)
4. Adaptive Public Relations (ref IUCN 2012, Section 8.1, Section 9

In addition, it is completely reasonable to highlight the importance of management structures for reintroductions, but seems unnecessary to bolster an argument for doing so by considering that previous reviews have neglected this aspect, failed to consider it, or made assumptions that biological factors are more important. Reviews from over 20 years ago are cited in support of this position, even though 20 years ago the whole field was relatively new and sensibly the focus was on fundamental biological aspects. It would not have made any sense to hone in on management structures while ignoring or downplaying absolutely critical aspects such as founder animal selection and release site habitat matching – these are and remain key drivers of success and no level of “political advocacy” or “adaptive outreach” will overcome problems arising because e.g. the original causes of extinction have not bee dealt with.

A more reasonable case could have been made by Sutton that, now that critical biological factors are being understood and examined, further gains could be made by looking at the features of management structures that best enable implementation of reintroduction projects that have a good grasp of the key biological aspects to be dealt with. But even here the case is pre-empted by the new IUCN Guidelines.

Additional comments

I have referenced these points by Line number on the submitted ms.

L5 A “reintroduction” is a type of “translocation” – it does not make sense to refer to them as either/or (ref IUCN 2012).

L35 Wording makes it seem that previous reviews of reintroduction outcomes have made assumptions about critical factors associated with success, but seminal quantitative reviews such as Griffith and the uncited Wolf et al papers, have been the basis for attention to key biological factors. It is incorrect therefore to refer to such analyses as “theories” (L32)

L51 Use standard terminology – do you mean “reinforcements” – reference the new (2012) IUCN Reintroduction Guidelines.

L55 Fischer and Lindenmayer study was not restricted to conservation translocations, but looked also at translocations for management of human wildlife conflict and game animal restocking. The cited success rate includes the very low success for translocations of problem animals and is not representative of reintroductions.

L82 Seem to be setting up a straw man in making an asserting that aspects such as e.g. publication bias, have gone largely unstudied. Even in a sense specific to reintroduction biology this is untrue (e.g. Bajomi 2010).

L88 Your thesis relates to organisational structures, and it is true that early reviews of reintroduction outcomes have not specifically related success to management, but it is not true to say that “human dimensions” were ignored. Consideration of the impacts of social acceptability and human actions, in relation to released animals, have been considered, e.g. even as far back as Kleiman 1989. Actually the importance of management structures for reintroductions has been highlighted by Beck (2001) in his review of the first reintroductions of bison in the US.

L95 You need to make a better case for your assertion that demography, genetics and ecology are not the “truly decisive factors “ – surely even the best management structure in the world could not overcome problems arising due to biological factors such as inappropriate habitat selection, inbreeding depression, predation, disease etc.

L98 You are trying too hard to criticise previous work – the quote is not “skeptical” it simply acknowledges a lack of data necessary to adequately assess management influence on reintroduction outcomes.

L108 Reading et al. provide you with a valuable strut to make your case, but instead you feel the need to criticise them for having “failed to follow through” with a more thorough examination. This critique is unnecessary to make your case, and an unwarranted attack on someone who provides you with essential support.

L149 Why did the early releases fail? What made these “experimental”?

L298 It is entirely unclear how it was determined that “four critical factors… contributed to … success” – this seems to be a qualitative judgement not well supported by any data presented. Also, you confound you own earlier case that biological factors are not decisive by here referring to the challenges relating to biological knowledge.

L309 With only a very small interviewee base you are restricted it seems to making a series of anecdotal points to support a priori assertions.

L319 On what basis can you claim that overcoming of challenges would have been unlikely?

L333 How can you separate out components of success in relation to autonomy of participants? This does not seem possible using the data presented because you cannot consider a counter-factual nor make comparisons with an independent project that did not succeed.

L339 Your data do not allow you to make conclusions about benefits to all “conservation practitioners”.

L344 Again, there is no basis for attributing success to this specific management factor.

L352 Hang on – is this selective citing – your introduction cite only Kleiman 1989 and part of a case that these factors have not been considered by past work. Here you reveal that Kleiman indeed highlighted the importance of internal evaluation.

·

Basic reporting

This is a well written and structured paper. The introduction and background information is good and positions the work within the wider academic context. All the figures are relevant to the content of the article.

Minor Correction - Line 388 ... (for an example of successful implementation of this strategy..) should read (UNSUCCESSFUL implementation ...)

Experimental design

The research question is clearly defined, relevant and topical. A historical case study approach, guided by grounded theory, is used to study the role of leadership and organisational culture in the successful Scottish sea eagle reintroduction project (1975-2012), The methods essentially consist of interviews along with archival research and the resultant data is both manually and digitally (Nvivo) analysed, all of which conform to standard methods.
There is some confusion in the Methods section (Line 170 on..) in relation to how many people were interviewed, and who were they? The number of interviewees varies from 11, 13, 15, 18 (lines 200-2007). I believe the actual number was 11 with four follow up interviews. The interviews form the core of the research and its findings, but a sample size of 11 project employees is very small. I believe we also need more information about who the interviewees are (without naming then), and the role they played in the project, All we are told is that they were full time employees of the project for at least three months, during any phase of the 37 year project (1975-2012). I suggest that the actual role of all the interviewees, [were they the Directors (Dennis & Love, present throughout the project), administrative or public relations staff, or biologists etc.], should be stated along with when and for how long they worked on the project. I recommend a table should be added with this information. Memory recall from a short term employee back in 1975 may not be very reliable!
The contribution of this research is in highlighting the importance of leadership and the organisational culture of the reintroduction project (a very valid point), but the author does not formally outline the organisational structure & decision making process of the sea eagle project. I recommend a flow chart/ section with this information. I also believe that the 'Interviewee Demographics' (Line 246- 255), should come under the Methods section of the paper, rather than the Results.
Finally the 'Interview Summary' section (Line 256-295) from which emerge the four main research findings is very short!
Line 320 - the author needs to explain more fully what 'Champion' style leadership is.
.

Validity of the findings

The paper makes an important and valid point - that species reintroduction is about more than just biology. The research findings relate to four key themes /recommendations for best practice organisational management in wildlife reintroduction programmes. The findings are valid, well made and have wider applicability beyond this case study research. But, as the author states in Line 397, - 'the findings are limited by their exploratory (& therefore preliminary) nature'. This research is valid but not substantial in nature, essentially due to the limited number of interviews and lack of clarity as to who was interviewed (as discussed above). It is exploratory work that requires more substantial research to fully validate its findings. However, it still makes a valid, if preliminary, contribution.

Additional comments

No commments

---

## Round 0.2 · Minor Revisions

Please have a look at the comments again and try revise as requested.
Regards, Michael

** Note from Staff to the author and reviewers: ** Due to the unavoidable situation of having multiple revisions and documents, there are several documents, all of which include (different) line numbering.Specifically: The authors submitted a word doc, and a tracked changes word doc (each of which contained line numbering, and each of which are available to the reviewers). Then our system creates a 'reviewing PDF' (which also contains line numbering and which reviewers tend to work from). And then, of course, there are multple 'revisions' of the submission. Therefore, we suggest that the author be explicit as to which version and document the line numbering is referring to.

·

Basic reporting

I cannot easily assess the validity of some responses where the author has chosen not to make changes because I cannot locate the specific text in the revised document as line numbers have changed. This has meant I cannot evaluate nor response to rebuttals. The authors need to indicate where sections now lies in the revised document.

I can highlight the following at this stage:

Original comment: L5 A “reintroduction” is a type of “translocation” – it does not make sense to refer to them as either/or (ref IUCN 2012).

Rebuttal: Not necessarily; reintroductions can be made wholly from a captive population – in this case, the animals are not being ‘translocated’ from one wild habitat to another, but rather released or reintroduced into any wild habitat at all.

My follow-up response: You are using the incorrect definition of “translocation” from the old guidelines, hence my reference to the current guidelines, where it is made clear that the origin of the animals to be released does not alter the type of translocation. Translocation is the overarching term, meaning movement (from anywhere) and release. Animals can therefore be reintroduced having been sourced from a wild population or a captive one.

Experimental design

Cannot comment - see note to Editor

Validity of the findings

Cannot comment - see note to Editor

---

## Round 0.3 · Minor Revisions

Dear Authors

Thanks for the revision. The MS is much improved. The referee thinks (and I agree) that you need to add a little on convincing the reader that the project has indeed been a success. I will not send the last revision back to him.

Regards

Michael Somers

·

Basic reporting

This is much improved and is now a useful contribution for conservation managers.

There is one key area needing a attention:
The ms proceeds from the basis of the presumed success of the sea eagle reintroduction, seeking to explore human and organisational factors that have contributed to "this success". It is essential therefore that you convince a reader that the project has indeed been a success. A section needs to be added to the review of the sea eagle programme, in which you define success as applied by the programme, list the relevant criteria applied, outline the outcome monitoring that was undertaken and the associated time frames, and make some quantified statement indicative of success.

Experimental design

No comments

Validity of the findings

No comments

---

## Round 0.4 · accepted · Accept

I have made some minor typo and font corrections on the attached pdf. Please sort these changes at final stage.